ecology

coral reefs, Scleractinia, seasonality, denitrification, dinitrogen fixation, diazotrophy

**Authors for correspondence:**
Arjen Tilstra
e-mail: tilstra@uni-bremen.de
Christian R. Voolstra
e-mail: chris.voolstra@gmail.com

# Relative abundance of nitrogen cycling microbes in coral holobionts reflects environmental nitrate availability

Arjen Tilstra[1], Florian Roth[2,3,4], Yusuf C. El-Khaled[1], Claudia Pogoreutz[2,5], Nils Rädecker[2,5,6], Christian R. Voolstra[2,5] and Christian Wild[1]

[1]Marine Ecology Department, Faculty of Biology and Chemistry, University of Bremen, Bremen, Germany
[2]Red Sea Research Center, King Abdullah University of Science and Technology, Thuwal, Kingdom of Saudi Arabia
[3]Baltic Sea Centre, Stockholm University, Stockholm, Sweden
[4]Tvärminne Zoological Station, Faculty of Biological and Environmental Sciences, University of Helsinki, Helsinki, Finland
[5]Department of Biology, University of Konstanz, Konstanz, Germany
[6]Laboratory for Biological Geochemistry, School of Architecture, Civil and Environmental Engineering, École Polytechnique Fédérale de Lausanne (EPFL), Lausanne, Switzerland

AT, 0000-0002-4337-6169; FR, 0000-0003-4004-5863;
YCE-K, 0000-0002-0401-5520; CP, 0000-0002-2853-7673;
NR, 0000-0002-2387-8567; CRV, 0000-0003-4555-3795;
CW, 0000-0001-9637-6536

Recent research suggests that nitrogen (N) cycling microbes are important for coral holobiont functioning. In particular, coral holobionts may acquire bioavailable N via prokaryotic dinitrogen ($N_2$) fixation or remove excess N via denitrification activity. However, our understanding of environmental drivers on these processes *in hospite* remains limited. Employing the strong seasonality of the central Red Sea, this study assessed the effects of environmental parameters on the proportional abundances of N cycling microbes associated with the hard corals *Acropora hemprichii* and *Stylophora pistillata*. Specifically, we quantified changes in the relative ratio between *nirS* and *nifH* gene copy numbers, as a proxy for seasonal shifts in denitrification and $N_2$ fixation potential in corals, respectively. In addition, we assessed coral tissue-associated Symbiodiniaceae cell densities and monitored environmental parameters to provide a holobiont and environmental context, respectively. While ratios of *nirS* to *nifH* gene copy numbers

varied between seasons, they revealed similar seasonal patterns in both coral species, with ratios closely following patterns in environmental nitrate availability. Symbiodiniaceae cell densities aligned with environmental nitrate availability, suggesting that the seasonal shifts in *nirS* to *nifH* gene abundance ratios were probably driven by nitrate availability in the coral holobiont. Thereby, our results suggest that N cycling in coral holobionts probably adjusts to environmental conditions by increasing and/or decreasing denitrification and $N_2$ fixation potential according to environmental nitrate availability. Microbial N cycling may, thus, extenuate the effects of changes in environmental nitrate availability on coral holobionts to support the maintenance of the coral–Symbiodiniaceae symbiosis.

# 1. Introduction

The oligotrophic nature of coral reefs requires an efficient use and recycling of the available nutrients within the ecosystem, including by their main engineers, scleractinian corals. As such, corals consist not only of the animal host alone but additionally harbour a diverse range of eukaryotic and prokaryotic microorganisms [1], rendering it a so-called 'holobiont'. Many of these coral-associated microorganisms aid in nutrient (re)cycling [2,3]. Nitrogen (N) is an essential macronutrient, the availability of which often being the controlling factor for primary production (i.e. the fixation of dissolved inorganic carbon (DIC) through photosynthesis performed by the Symbiodiniaceae) in coral holobionts [4,5]. Despite the importance of N for coral holobionts, *in hospite* limitation of N is crucial for maintaining the symbiosis between the coral animal and the photosynthetic algal symbionts of the family Symbiodiniaceae [6]. Translocation of photosynthates by Symbiodiniaceae, i.e. the main supply of organic C for the coral host [7], is optimal under N limitation [8–10]. The interruption of N limitation may, thus, lead to the cessation of photosynthate translocation, which may ultimately lead to the breakdown of the coral–Symbiodiniaceae symbiosis due to increased bleaching susceptibility [9,11,12]. Thus, the cycling of N is critical for understanding coral holobiont functioning [9].

The environmental availability of N fluctuates in coral reef environments. This may include natural fluctuations, e.g. seasonality in N availability [13–16], as well as anthropogenic N inputs [17]. In this sense, coral-associated microbes, in particular prokaryotes, may play an integral role in coral holobiont N cycling. On the one hand, diazotrophs, prokaryotes capable of fixing atmospheric dinitrogen ($N_2$), may provide the coral holobiont with de novo bioavailable N in the form of ammonium in times of environmental N scarcity [18–20]. On the other hand, microbes capable of denitrification, i.e. the chemical reduction of nitrate to $N_2$, may play a putative role in alleviating the coral holobiont from excess N [21,22]. It was hypothesized that high denitrification rates may maintain N limitation for Symbiodiniaceae, and, as a result, may potentially support the functioning of the coral–Symbiodiniaceae symbiosis [9,11]. To this end, the presence of denitrifiers in coral holobionts was first reported in the late 2000s [23,24], but Tilstra *et al.* [21] only recently demonstrated that denitrification indeed constitutes an active metabolic pathway present in coral holobionts from the oligotrophic central Red Sea.

Taken together, microbial N cycling contributes to N availability for the coral holobiont. However, our understanding of how abiotic and biotic factors affect N cycling properties in corals remains poorly understood. Making use of the pronounced seasonality of the central Red Sea, the present study aimed to (i) assess patterns in the abundance of denitrifiers (approximated via *nirS* gene copy numbers) in relation to diazotrophs (approximated via *nifH* gene copy numbers) in a seasonal resolution (herein referred to as *nirS* to *nifH* gene abundance ratios); and (ii) identify environmental parameters potentially driving the observed seasonal patterns. Due to the potential stimulating or suppressing effects of dissolved inorganic N (DIN) on denitrification [22,25] and diazotrophy [22,26–29], respectively, we hypothesized that the seasonal patterns of *nirS* to *nifH* gene abundance ratios in coral holobionts would be mostly affected by DIN, i.e. nitrate, nitrite and/or ammonium concentrations.

# 2. Material and methods

## 2.1. Sample collection

Two common species of hard coral (figure 1*a*), i.e. *Acropora hemprichii* (Acroporidae) and *Stylophora pistillata* (Pocilloporidae), were collected over four seasons: April 2017 (spring), August 2017

(a)

*Acropora hemprichii*    *Stylophora pistillata*

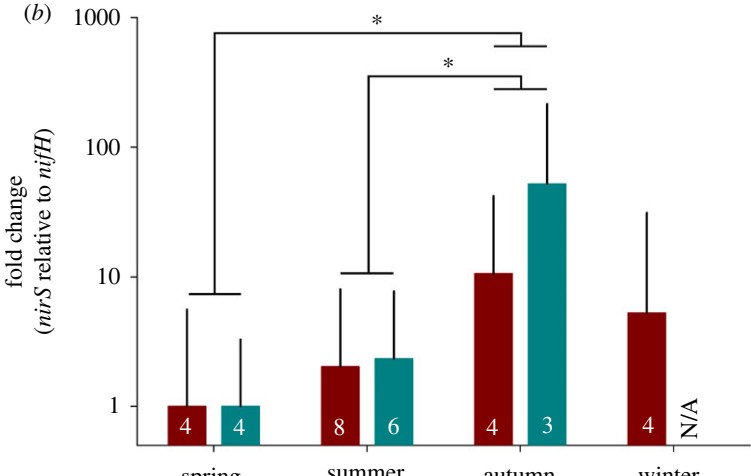

(b)

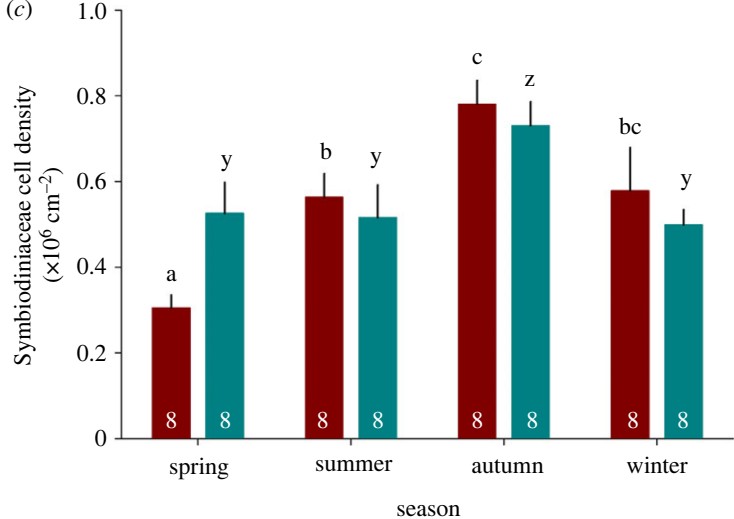

(c)

**Figure 1.** Patterns of *nirS* to *nifH* gene abundance ratios and Symbiodiniaceae cell densities associated with two central Red Sea hard coral species across four seasons. (a) Representative photographs of investigated species, (b) fold change of *nirS* to *nifH* gene abundance ratios and (c) Symbiodiniaceae cell densities. Fold changes were calculated in relation to spring, during which both species exhibited the lowest *nirS* to *nifH* gene abundance ratios; bars indicate the mean; error bars indicate upper confidence intervals (+1 s.e.). Numbers in the bars represent the sample size (*n*). Within plot (b), asterisks indicate significant differences between seasons (pair-wise PERMANOVA, *$p < 0.05$). Within plot (c), different letters above error bars indicate statistically significant differences per species between seasons (pair-wise PERMANOVA, $p < 0.05$). N/A = not available.

(summer), November 2017 (autumn) and January 2018 (winter). Corals were collected at approx. 5 m water depth at the semi-exposed side of the inshore reef Abu Shosha (22°18′15″ N, 39°02′56″ E) located in the Saudi Arabian central Red Sea. Sailing permits were issued by the Saudi Arabian Coastguard Authority to the sites that included coral collection. During each season, eight fragments of each coral species were collected from spatially separated colonies (distance apart greater than 10 m) to ensure genetic diversity. Immediately after collection, fragments were flash frozen in liquid N aboard the

**Table 1.** Selected primers used for amplification.

| target gene | primer | nucleotide sequence (5′ → 3′) | reference |
|---|---|---|---|
| *nirS* | cd3aF | GTSAACGTSAAGGARACSGG | [31] |
| | R3cd | GASTTCGGRTGSGTCTTGA | |
| *nifH* | F2 | TGYGAYCCIAAIGCIGA | [32] |
| | R6 | TCIGGIGARATGATGGC | |

research vessel. Subsequently, fragments were transported to the laboratories of the King Abdullah University of Science and Technology and stored at −80°C until further processing.

## 2.2. DNA extraction and quantitative PCR

Coral tissues were separated from the skeleton by pressurized air. DNA was extracted from 100 µl of the resulting tissue slurry using the Qiagen DNeasy Plant Mini Kit (Qiagen, Germany) according to the manufacturer's instructions. Extracted DNA was stored at −20°C until qPCR assays were performed. Quantitative PCRs (qPCRs) were carried out according to Tilstra *et al.* [21]. Briefly, relative copy numbers of the functional genes *nirS* and *nifH* were used as a proxy for denitrification and diazotrophy, respectively, as implemented previously [20,21,30]. qPCR assays were performed in technical triplicates for each biological replicate (i.e. coral fragment), which were averaged before statistical analysis. Each assay contained 9 µl reaction mixture and 1 µl DNA template (input adjusted to approx. 3 ng DNA µl$^{-1}$). The reaction mixture contained 5 µl Platinum SYBR Green qPCR Master Mix (Invitrogen, Carlsbad, CA, United States), 0.2 µl of each primer (10 µM), 0.2 µl of ROX dye and 3.4 µl of nuclease-free water (see table 1 for primers used). *NirS* to *nifH* gene abundance ratios were determined by normalizing against the *nifH* gene. The thermal cycling protocol was 50°C for 2 min, 95°C for 2 min, 50 cycles of 95°C for 30 s, 51°C for 1 min, 72°C for 1 min and a 72°C extension cycle for 2 min. Amplification specificity was determined by adding a dissociation step (melting curve analysis). All assays were performed on the ABI 7900HT Fast real-time PCR System (Applied Biosystems, CA, USA). Standard calibration curves were run simultaneously covering eight orders of magnitude ($10^1$–$10^8$ copies of template per assay each for the *nirS* and *nifH* gene). The qPCR efficiency (E) of the primer pairs was 86% and 87%, respectively, calculated according to the equation $E = [10^{(-1/\text{slope})} - 1]$. *nirS* to *nifH* gene abundance ratios were calculated as $2^{(-\Delta\Delta Ct)}$ against *nifH* Ct values using the season with the lowest relative abundances as the reference [33].

## 2.3. Symbiodiniaceae cell densities

A further aliquot of the tissue slurry used for DNA extraction was used to obtain cell densities of Symbiodiniaceae. Tissue slurry aliquots were homogenized, diluted at a ratio of 3 : 1 or 5 : 1, and Symbiodiniaceae cells were subsequently counted using a Neubauer-improved haemocytometer on a light microscope with HD camera (Zeiss, Germany). Resulting photographs were analysed using the Cell Counter Notice in ImageJ software (National Institutes of Health, USA). Cell counts for tissue slurries for each individual coral were done in duplicates and subsequently averaged. Finally, to obtain cell densities of Symbiodiniaceae per unit area of coral tissue, cell counts were normalized to the coral surface area, which was calculated using cloud-based three-dimensional models of samples (Autodesk Remake v. 19.1.1.2) [34,35].

## 2.4. Environmental parameters

Environmental data, i.e. temperature, light intensity (photosynthetically active radiation [PAR = 400–700 nm], salinity, dissolved oxygen (DO), nitrate, nitrite, ammonium, dissolved inorganic phosphorus (DIP = [phosphate]) and dissolved organic carbon (DOC), were described and published previously in Roth *et al.* [14,36] and were reanalysed for the purposes of the present study.

The temperature was measured continuously with data loggers (Onset HOBO Water Temperature Pro v. 2 Data Logger U22–001; accuracy: ± 0.21°C) one month prior and within the month of sampling on a 30 min interval. Light availability (lux) was measured with data loggers (Onset HOBO Pendant

UA-002-64; spectral detection range 150–1200 nm) for 3 full days every month and converted to photosynthetically active radiation (PAR = 400–700 nm) using a conversion factor of 51.8 as outlined by Roth *et al*. [14]. The conversion factor was obtained by inter-calibrating the lux readings (i.e. from the Onset HOBO Pendant) with data obtained from a parallel deployment of a PAR sensor (LI-COR LI-1500 quantum sensor) during 4 h of daylight. Both readings correlated ($r^2 = 0.91$) and the obtained conversion factor was 51.8. Salinity was measured on three days every month using a conductivity measuring cell (TetraCon®, 925, WTW, accuracy: ± 0.5% of value, internal conversion to salinity). DO was quantified on 2 days within the month of sampling by taking the average of eight autonomous recording DO and temperature sensors (HOBO U26; temperature corrected and salinity adjusted) that were deployed at 5 m water depth within a radius of 50 m of the sampling site. Seawater samples were taken in triplicates from directly above the reef at the sampling site on 3 days monthly or bimonthly, i.e. one month prior to sampling and/or the month of sampling, to measure (in)organic nutrients. Nitrate, nitrite and DIP were measured photometrically, while ammonium was measured fluorometrically. DIN was calculated as [nitrate] + [nitrite] + [ammonium]. Subsamples for DOC were filtered through 0.2 µm Millipore® polycarbonate filters into pre-combusted (450°C, 4.5 h) acid-washed amber glass vials (Wheaton) with Teflon-lined lids, and samples were subsequently acidified with $H_3PO_4$ until reaching pH 1–2. Samples were kept in the dark at 4°C until further analysis by high-temperature catalytic oxidation using a total organic carbon analyser (Shimadzu, TOC-L). To monitor the accuracy of DOC concentration measurements, we used reference material of deep-sea carbon ($42$–$45$ µmol C $l^{-1}$) and low-carbon water ($1$–$2$ µmol C $l^{-1}$).

## 2.5. Statistical analyses

To assess seasonality, data were analysed using non-parametric permutational multivariate analysis of variance (PERMANOVA) using PRIMER-E version 6 software [37] with the PERMANOVA + add on [38]. To test for differences in *nirS* and *nifH* gene abundance ratios and Symbiodiniaceae cell densities between seasons, two-factorial univariate PERMANOVAs were performed with season and coral species as main factors based on Euclidian distances of square-root transformed data [39], while one-factorial univariate PERMANOVAs were performed with the season as the main factor for environmental parameters based on Euclidian distances of normalized data [39]. Type III (partial) sum of squares was therefore used with an unrestricted permutation of raw data (999 permutations), and PERMANOVA pair-wise tests with parallel Monte Carlo tests (with Bonferroni corrected *p*-values) were carried out when significant differences were found, to account for multiple comparisons.

Pearson product-moment correlation tests were performed to identify correlations between *nirS* to *nifH* gene abundance ratios, Symbiodiniaceae cell density and environmental variables. Salinity and DIP were omitted from the analyses as differences were assumed to have no ecological significance. In addition, nitrite and ammonium were omitted from the analyses as there were no significant differences between seasons (electronic supplementary material, table S1). Finally, linear regression analysis was used to assess a potential statistical relationship between *nirS* to *nifH* gene abundance ratios and Symbiodiniaceae cell density over all seasons. All values are given as mean ± s.e.

# 3. Results

## 3.1. *nirS* to *nifH* gene abundance ratios

The extracted DNA was of varying quality, and amplification was not possible in some samples, resulting in varying levels of replicates for each species and season. The resulting sample size is indicated within figure 1*b*.

Lowest *nirS* (as a proxy for denitrification) to *nifH* (as a proxy for $N_2$ fixation) gene abundance ratios were observed during the spring season, hence gene abundance ratios for other seasons were calculated as fold changes in relation to spring (figure 1*b*). In summer, gene abundance ratios increased approximately twofold for both species, while they increased approximately 11-fold during autumn for *A. hemprichii* and approximately 52-fold for *S. pistillata* (figure 1*b*). During winter, gene abundance ratios were approximately fivefold higher in *A. hemprichii* compared with spring (no data available for *S. pistillata* for this season) (figure 1*b*). There was no interactive effect of season and species on gene abundance ratios (PERMANOVA, pseudo-*F* = 0.13, *p* = 0.898; electronic supplementary material, table S1). However, there was an effect of season (PERMANOVA, pseudo-*F* = 3.27, *p* = 0.039; electronic

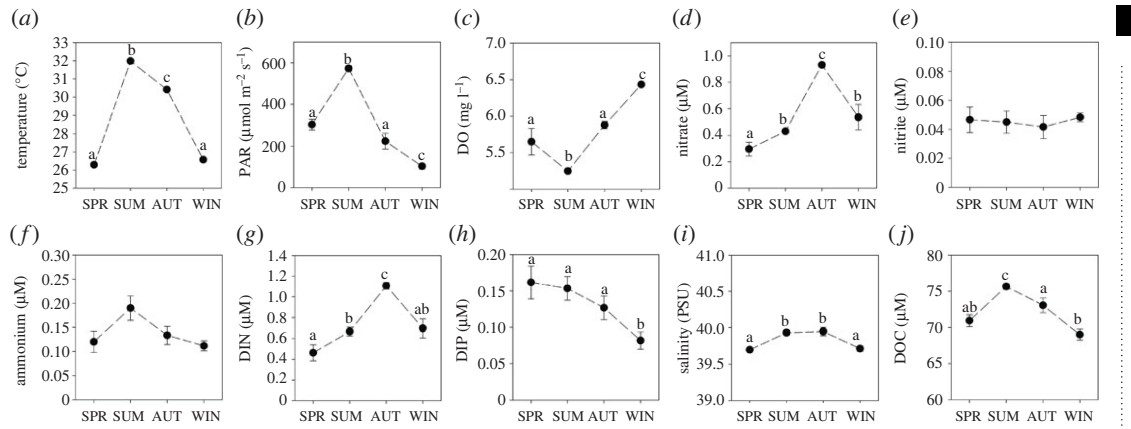

**Figure 2.** Means (±s.e.) of environmental parameters measured over four seasons. (*a*) Temperature, (*b*) PAR, (*c*) DO, (*d*) nitrate, (*e*) nitrite, (*f*) ammonium, (*g*) dissolved inorganic nitrogen (DIN = [nitrate] + [nitrite] + [ammonium]), (*h*) dissolved inorganic phosphorus (DIP = [phosphate]), (*i*) salinity and (*j*) DOC. Different letters above error bars indicate significant differences between seasons within each plot ($p < 0.05$). SPR = spring; SUM = summer; AUT = autumn; WIN = winter. Data were extracted from Roth *et al.* [14,36] and re-analysed for the purpose of this study.

supplementary material, table S1) and species (PERMANOVA, pseudo-$F = 6.22$, $p = 0.022$; electronic supplementary material, table S1) on gene abundance ratios. Indeed, gene abundance ratios were higher during autumn compared with spring (pair-wise PERMANOVA, $t = 3.22$, $p = 0.015$) and summer (pair-wise PERMANOVA, $t = 2.67$, $p = 0.015$). See electronic supplementary material, table S2 for full details of pair-wise comparisons.

## 3.2. Symbiodiniaceae cell densities

Cell densities of Symbiodiniaceae varied more strongly between seasons in *A. hemprichii* compared with *S. pistillata* (electronic supplementary material, table S1; figure 1*c*). Cell densities for *A. hemprichii* were lowest in spring ($0.31 \pm 0.03 \times 10^6$ cells cm$^{-2}$) and significantly increased during summer ($0.56 \pm 0.05 \times 10^6$ cells cm$^{-2}$; pair-wise PERMANOVA, $t = 4.29$, $p = 0.001$) (figure 1*c*). Subsequently, cell densities significantly increased in autumn ($0.78 \pm 0.06 \times 10^6$ cells cm$^{-2}$; pair-wise PERMANOVA, $t = 2.78$, $p = 0.021$), but returned to densities similar to summer, during winter ($0.59 \pm 0.10 \times 10^6$ cells cm$^{-2}$) (figure 1*c*). Cell densities of Symbiodiniaceae in tissues of *S. pistillata* were similar during spring, summer and winter ($0.53 \pm 0.07$, $0.52 \pm 0.08$, $0.50 \pm 0.04 \times 10^6$ cells cm$^{-2}$, respectively) (figure 1*c*). However, densities during autumn were significantly higher compared with the other seasons ($0.73 \pm 0.06 \times 10^6$ cells cm$^{-2}$; pair-wise PERMANOVA, $p < 0.05$) (figure 1*c*). See electronic supplementary material, table S2 for full details of pair-wise comparisons.

## 3.3. Environmental parameters

Several environmental parameters exhibited marked seasonal fluctuations (figure 2; electronic supplementary material, table S1). Temperature and PAR increased from spring to the summer season when both parameters were at their highest ($31.99 \pm 0.01$°C and $573 \pm 13$ µmol m$^{-2}$ s$^{-1}$, respectively) (figure 2*a* and *b*). DO was lowest in summer ($5.25 \pm 0.03$ mg l$^{-1}$) and highest in winter ($6.44 \pm 0.03$ mg l$^{-1}$) (figure 2*c*). Nitrate was highest during the autumn season ($0.93 \pm 0.02$ µM) and lowest during the spring season ($0.30 \pm 0.05$ µM) (figure 2*d*). Nitrite remained stable throughout all seasons ($0.05 \pm 0.01$ µM) (figure 2*e*). Ammonium remained stable throughout all seasons ($0.14 \pm 0.02$ µM) (figure 2*f*). DIN followed the same pattern as nitrate being highest during the autumn season ($1.11 \pm 0.03$ µM) and lowest during the spring season ($0.46 \pm 0.08$ µM) (figure 2*g*; electronic supplementary material, figure S1). DIP was stable from spring until autumn but decreased during winter ($0.08 \pm 0.01$ µM) (figure 2*h*). Salinity remained relatively stable throughout the period of study ($39.85 \pm 0.02$ PSU) (figure 2*i*). DOC was highest in summer ($75.66 \pm 0.40$ µM) and lowest in spring ($70.93 \pm 0.82$ µM) and winter ($68.99 \pm 0.77$ µM) (figure 2*j*). See electronic supplementary material, table S2 for full details of pair-wise comparisons.

**Figure 3.** Pearson product-moment correlation analyses for (*a*) ΔCt of *nirS-nifH* against environmental nitrate concentrations pooled for both coral species, cell densities of Symbiodiniaceae against environmental nitrate concentrations for (*b*) *A. hemprichii* and (*c*) *S. pistillata* and linear regression analyses for (*d*) ΔCt of *nirS-nifH* against cell densities of Symbiodiniaceae pooled for both coral species, for (*e*) *A. hemprichii* and (*f*) *S. pistillata*. *r* = Pearson coefficient. Best-fit linear regression lines ± 95% confidence intervals (dotted lines) are solid when a significant relationship was established; lines are dashed when not significant.

## 3.4. Correlation analyses

Due to the lack of a significant interaction between season and species in *nirS* to *nifH* gene abundance ratios (electronic supplementary material, table S1), data for both species were pooled for correlation analyses.

The most significant correlation for both species' *nirS* to *nifH* gene abundance ratios was with nitrate (Pearson product-moment correlation, $r = 0.463$, $p = 0.007$; figure 3*a*; electronic supplementary material, table S3). Symbiodiniaceae cell densities also correlated most significantly with nitrate for both *A. hemprichii* (Pearson product-moment correlation, $r = 0.649$, $p = 0.001$; figure 3*b*; electronic supplementary material, table S3) and *S. pistillata* (Pearson product-moment correlation, $r = 0.446$, $p = 0.011$; figure 3*c*; electronic supplementary material, table S3).

No relationships were found between *nirS* to *nifH* gene abundance ratios and Symbiodiniaceae cell densities for both corals combined (linear regression, $F = 2.61$, $r = 0.279$, $p = 0.116$; figure 3*d*), or per single species; *A. hemprichii* (linear regression, $F = 1.35$, $r = 0.264$, $p = 0.260$; figure 3*e*) and *S. pistillata* (linear regression, $F = 2.21$, $r = 0.409$, $p = 0.165$; figure 3*f*).

## 4. Discussion

Coral-associated microbial N cycling still remains an understudied, but arguably very important aspect of coral holobiont functioning as it may be a source or sink of bioavailable N [9,40]. Here, we assessed the proportional dynamics of two antagonistic N cycling pathways, i.e. denitrification and diazotrophy, via means of the relative abundance of key genes, in two common central Red Sea coral species (figure 1*a*) in a seasonal resolution. To this end, proportional abundances of the functional marker gene *nirS*, as a proxy for denitrification [21], were calculated in relation to the functional marker gene *nifH*, as a proxy for diazotrophy [20] (figure 1*b*). Importantly, the *nirS* to *nifH* gene abundance ratios presented

in this study are not based on absolute, but relative, abundances of each respective marker gene. Consequently, this approach does not allow for any conclusion regarding the absolute abundance of marker genes, or, ultimately, absolute abundances of either denitrifiers or diazotrophs. Rather, changes in the ratio may be interpreted as a proxy for a shift in the relative abundance of denitrifying in relation to $N_2$-fixing prokaryotes and serve as a proxy for the relative prevalence of the associated processes/pathways [20,21]. In this light, increasing ratios may reflect an increase in denitrifying microbes and/or a decrease in $N_2$-fixing microbes and vice versa. Using this approach, we were able to characterize the seasonal dynamics of microbial N cycling in central Red Sea corals.

## 4.1. Seasonal patterns and environmental drivers of denitrification and $N_2$ fixation potential in corals

The *nirS* to *nifH* gene abundance ratios followed a very similar pattern in both coral species across seasons (figure 1*b*). Autumn was characterized by the highest and spring by the lowest *nirS* to *nifH* gene abundance ratios, regardless of species. While relative gene abundances of both marker genes (i.e. *nirS* and *nifH*) do not allow any direct conclusion regarding activities of associated biological processes, previous studies using corals from the same location showed that relative abundances of marker genes correlated with denitrification and $N_2$ fixation rates under these environmental conditions [20,21]. Consequently, the observed patterns of *nirS* to *nifH* gene abundance ratios may translate into similar seasonal patterns for associated denitrification to $N_2$ fixation activities. In this light, the similarity of the seasonal patterns found in both coral species suggests that the functional niche occupied by different N cycling microbes may be very similar and highly responsive to changing environmental conditions.

Among all investigated environmental parameters, nitrate (the most significant contributor to DIN throughout all four investigated seasons; electronic supplementary material, figure S1) and DIN concentrations showed a moderate correlation with relative *nirS* to *nifH* gene abundance ratios across coral species (electronic supplementary material, table S3). As the substrate for denitrifiers, nitrate may directly stimulate denitrification activity [22,25]. Likewise, increased nitrate and/or ammonium concentrations have been shown to depress diazotroph activity [26–29,41]. The observed patterns in relative *nirS* to *nifH* gene abundance ratios may, thus, be the direct consequence of increased environmental N availability in the coral holobiont.

The notion of seasonally changing N availability driving patterns in ratios of prokaryotic N cycling functional groups within coral holobionts is corroborated by the patterns of Symbiodiniaceae cell densities observed in both coral species. Similar to *nirS* to *nifH* gene abundance ratios, Symbiodiniaceae cell densities exhibited strong seasonal differences, comparable to Symbiodiniaceae cell densities of conspecifics originating from the Red Sea [21,40,42,43], that positively correlated with environmental nitrate and DIN concentrations (figure 3*b,c*; electronic supplementary material, table S3). As the coral holobionts' internal N concentrations were not measured in the present study, it is unknown whether environmental N was reflected by bioavailable N availability within the holobionts. However, Symbiodiniaceae population densities are known to be governed by N availability in the stable coral–algae symbiosis [44–46], suggesting that environmental N availability was closely linked with N availability within the coral holobiont in the present study as previously observed in *ex situ* studies [47–49].

## 4.2. Dynamics of N cycling microbes as a buffer against seasonal changes in environmental N availability

Limited N availability is critical to coral holobiont functioning as it limits population growth of Symbiodiniaceae *in hospite* and maintains high rates of translocation of photosynthetic carbon (C) to the host [8–10,46]. Seasonal or anthropogenically driven increases in environmental N availability may consequently stimulate Symbiodiniaceae proliferation, thereby disrupting or reducing organic C translocation to the host, ultimately posing a threat to overall coral holobiont functioning [11,50]. Yet, coral holobionts manage to thrive in highly dynamic environments with considerable temporal and spatial variations in N availability [14,15,22,51]. The positive correlation of Symbiodiniaceae densities and environmental N availability in the present study suggests that the coral hosts may not have been able to fully maintain stable N availability within the holobiont. In this light, the shift in relative *nirS* to *nifH* gene abundance ratios in response to the availability of environmental N could prove

beneficial to the coral holobiont by partly stabilizing N levels in the host relative to environmental fluctuations. As such, during periods of low N availability (e.g. spring), low relative *nirS* to *nifH* gene abundance ratios probably imply reduced denitrification and increased $N_2$ fixation activity. Likewise, during periods of high N availability (e.g. autumn), high relative *nirS* to *nifH* gene abundance ratios probably reflect increased denitrification and reduced $N_2$ fixation activity. If indeed translatable to corresponding prokaryotic activity, the observed dynamics in functional N cycling gene abundance ratios may, thus, directly support coral holobiont functioning [9]. Specifically, the interplay of denitrifiers and $N_2$-fixers may support the removal of excess N, while providing access to new bioavailable N in times of low environmental N availability [9,18]. While these processes may evidently have been insufficient for the stabilization of N availability within the holobiont, as shown in the present study, they may be directly assisting the coral host in regulating Symbiodiniaceae populations.

## 4.3. Future research directions

The present study adds to a rapidly growing body of research, highlighting the functional importance of N cycling microbes in coral holobiont functioning. Deciphering the interactions between N cyclers, not just those capable of $N_2$ fixation and denitrification but also potentially important N cyclers capable of e.g. nitrification and ANAMMOX, and other coral holobiont members promises to advance our understanding of coral holobiont functioning in light of environmental conditions and anthropogenically driven change. While combined molecular (sequencing, real-time PCR) and physiological approaches haven proven powerful tools to study N cycling properties of coral holobionts, future studies should aim to address the identification and localization of the main microbial players along with accurate quantification of metabolic interactions with other holobiont members. In this light, fluorescence *in situ* hybridization, as well as nanoscale secondary ion mass spectrometry (NanoSIMS) techniques, may allow for an integrated and functional understanding of metabolic interactions in light of their localization within the coral holobiont.

Ethics. Corals were collected adhering to local guidelines for sampling and treatment. Sailing permits were issued by the Saudi Arabian Coastguard Authority to the sites that included coral collection.

Data accessibility. The supporting data are provided in the electronic supplementary material and in the Dryad Digital Repository: https://doi.org/10.5061/dryad.rjdfn2z8k [52].

Authors' contributions. All authors conceived and designed the experiment. A.T., F.R., Y.C.E.-K. and N.R. carried out field sampling. A.T., F.R. and Y.C.E.-K. carried out molecular, physiological and/or chemical measurements. A.T., F.R. and N.R. performed data and/or statistical analyses. A.T. wrote the manuscript. All authors revised drafts of the manuscript.

Competing interests. We declare we have no competing interests.

Funding. Financial support was provided by KAUST baseline funds to C.R. Voolstra and the German Research Foundation (DFG) grant nos. Wi 2677/9-1 and Wi 2677/16-1 to C.W.

Acknowledgements. We thank KAUST CMOR staff and boat crews for their support with diving operations. We thank Nauras Daraghmeh for his support with re-analysing environmental parameter data.

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
