## [Peer Review File · Royal Society Open Science]

Review History

RSOS-201835.R0 (Original submission)

Review form: Reviewer 1 (Tim Wijgerde)

Is the manuscript scientifically sound in its present form?

Yes

Are the interpretations and conclusions justified by the results?

Yes

Is the language acceptable?

Yes

Do you have any ethical concerns with this paper?

No

Have you any concerns about statistical analyses in this paper?

Yes

Recommendation?

Accept with minor revision (please list in comments)

Comments to the Author(s)

The authors have presented an insightful and well-written manuscript which is almost ready for publication. I only have a list of minor issues, which mostly concern the statistical part of the paper.

Methods:

Line 152: what is this conversion factor based on? I do not see sources. Is this based on the spectral light quality in situ?

Results:

Supplementary Table S1 lists no interactive effect of species and season on gene ratios. Thus, the authors can only resort to testing differences between seasons for both species combined (i.e. averaging the data from both species for each season). However, as shown in figure 1B, the authors have tested the means of the different seasons against each other per species. Splitting the data into both factors and doing pairwise comparisons like this can only be justified when the PERMANOVA first yields a significant interaction. I recommend testing the various seasons against one another without splitting the data into both species (i.e. performing a post-hoc test to follow up the significant main effect of season). Indeed, this is what the authors have done and described in lines 205 to 207. I would therefore change the pairwise comparisons in figure 1 to reflect these lines. Lines 194 to 198 should be removed as these suggest an interactive effect, which is not there. I would first report significant effects and then report the ratios which significantly differ, with species pooled. Supplementary table S2 should not list parts A and B, but should include C, where the species have been correctly pooled (and the rest of the table, of course). Interestingly, the authors did pool species for their correlations as described in lines 259 to 261. I agree with this approach, by the way. I hope the pairwise tests account for inflation of the familywise error (i.e. the p-values were corrected to compensate for multiple comparisons. Was this why you did parallel Monte Carlo tests)? I would make this issue clear in the M and M, because p values can be adjusted or a Bonferroni correction (with α / number of pairwise tests) can be done. In relation to this: Figure 1C is correctly done as there is a significant interactive effect for cell densities.

Table S2 details pairwise comparisons for environmental variables (f to o), but I do not see a main PERMANOVA table with season as factor and a corresponding pseudo-F, which should be presented before performing pairwise (post-hoc) comparisons, correct? If so, table S1 should be expanded accordingly. Indeed, line 174 does mention one-way PERMANOVA's but these are not listed in table S2.

The axes of figure 2 are hard to read; can the font size be increased?

Table S3 has a colour legend, it seems easier to read if you just write down the actual r values with asterisks for significance. Perhaps the editor can decide on this?

The r values shown in figure 3 and table s3 are significant, but only weak to moderate. I would reflect this in the text. Now, lines 262 to 264 could suggest to the reader that strong correlations have been found, which is not the case.

Discussion:

Line 301: ..., regardless of species (there was no interactive effect).

Line 311: ...showed a moderate correlation...

I really like the discussion; perhaps the authors could also briefly touch upon the zoox densities found per se? The highest values, around 1 million zoox per cm², are what we commonly find in healthy corals growing in aquaria (where nutrient levels are quite higher than in the field). I am unsure about common densities found in the field, but perhaps it would be interesting to briefly

mention something about other field data from the literature. Are these densities in a typical range for the field?

Review form: Reviewer 2

Is the manuscript scientifically sound in its present form?

Yes

Are the interpretations and conclusions justified by the results?

No

Is the language acceptable?

Yes

Do you have any ethical concerns with this paper?

No

Have you any concerns about statistical analyses in this paper?

Yes

Recommendation?

Major revision is needed (please make suggestions in comments)

Comments to the Author(s)

See attached file (Appendix A).

Decision letter (RSOS-201835.R0)

Dear Dr Tilstra

The Editors assigned to your paper RSOS-201835 "Relative abundance of nitrogen cycling microbes in coral holobionts reflects environmental nitrate availability" have now received comments from reviewers and would like you to revise the paper in accordance with the reviewer comments and any comments from the Editors. Please note this decision does not guarantee eventual acceptance.

We do not generally allow multiple rounds of revision so we urge you to make every effort to fully address all of the comments at this stage. If deemed necessary by the Editors, your

manuscript will be sent back to one or more of the original reviewers for assessment. If the original reviewers are not available, we may invite new reviewers.

Please submit your revised manuscript and required files (see below) no later than 21 days from today's (ie 19-Feb-2021) date. Note: the ScholarOne system will 'lock' if submission of the revision is attempted 21 or more days after the deadline. If you do not think you will be able to meet this deadline please contact the editorial office immediately.

on behalf of Professor Pete Smith (Subject Editor)
openscience@royalsociety.org

Associate Editor Comments to Author;

Please accept our apologies for the unusual delay in getting to this decision stage - unfortunately, we've found that, for a great many papers, it is proving more difficult to engage reviewers' support (for entirely understandable reasons, given the COVID crisis), we nevertheless understand it is frustrating for authors to receive a decision several months after initially submitting. In any case, two referees have submitted a report on your work - they each recommend a number of changes (some comparatively straightforward, others may require a little more engagement), and on this basis, we're offering you the opportunity to revise the manuscript.

Reviewer comments to Author:

Reviewer: 1
Comments to the Author(s)

The authors have presented an insightful and well-written manuscript which is almost ready for publication. I only have a list of minor issues, which mostly concern the statistical part of the paper.

Methods:

Line 152: what is this conversion factor based on? I do not see sources. Is this based on the spectral light quality in situ?

Results:

Supplementary Table S1 lists no interactive effect of species and season on gene ratios. Thus, the authors can only resort to testing differences between seasons for both species combined (i.e. averaging the data from both species for each season). However, as shown in figure 1B, the authors have tested the means of the different seasons against each other per species. Splitting the data into both factors and doing pairwise comparisons like this can only be justified when the PERMANOVA first yields a significant interaction. I recommend testing the various seasons against one another without splitting the data into both species (i.e. performing a post-hoc test to follow up the significant main effect of season). Indeed, this is what the authors have done and described in lines 205 to 207. I would therefore change the pairwise comparisons in figure 1 to reflect these lines. Lines 194 to 198 should be removed as these suggest an interactive effect, which is not there. I would first report significant effects and then report the ratios which significantly differ, with species pooled. Supplementary table S2 should not list parts A and B, but should include C, where the species have been correctly pooled (and the rest of the table, of course). Interestingly, the authors did pool species for their correlations as described in lines 259 to 261. I agree with this approach, by the way. I hope the pairwise tests account for inflation of the familywise error (i.e. the p-values were corrected to compensate for multiple comparisons. Was this why you did parallel Monte Carlo tests)? I would make this issue clear in the M and M, because p values can be adjusted or a Bonferroni correction (with $\alpha / \text{number of pairwise tests}$) can be done. In relation to this: Figure 1C is correctly done as there is a significant interactive effect for cell densities.

Table S2 details pairwise comparisons for environmental variables (f to o), but I do not see a main PERMANOVA table with season as factor and a corresponding pseudo-F, which should be presented before performing pairwise (post-hoc) comparisons, correct? If so, table S1 should be expanded accordingly. Indeed, line 174 does mention one-way PERMANOVA's but these are not listed in table S2.

The axes of figure 2 are hard to read; can the font size be increased?

Table S3 has a colour legend, it seems easier to read if you just write down the actual r values with asterisks for significance. Perhaps the editor can decide on this?

The r values shown in figure 3 and table S3 are significant, but only weak to moderate. I would reflect this in the text. Now, lines 262 to 264 could suggest to the reader that strong correlations have been found, which is not the case.

Discussion:

Line 301: ..., regardless of species (there was no interactive effect).

Line 311: ...showed a moderate correlation...

I really like the discussion; perhaps the authors could also briefly touch upon the zoox densities found per se? The highest values, around 1 million zoox per cm², are what we commonly find in healthy corals growing in aquaria (where nutrient levels are quite higher than in the field). I am unsure about common densities found in the field, but perhaps it would be interesting to briefly mention something about other field data from the literature. Are these densities in a typical range for the field?

Reviewer: 2

Comments to the Author(s)

See attached file.

===PREPARING YOUR MANUSCRIPT===

===PREPARING YOUR REVISION IN SCHOLARONE===

- Any electronic supplementary material (ESM).
- If you are requesting a discretionary waiver for the article processing charge, the waiver form must be included at this step.
- If you are providing image files for potential cover images, please upload these at this step, and inform the editorial office you have done so. You must hold the copyright to any image provided.
- A copy of your point-by-point response to referees and Editors. This will expedite the preparation of your proof.

- Ensure that your data access statement meets the requirements at <https://royalsociety.org/journals/authors/author-guidelines/#data>. You should ensure that you cite the dataset in your reference list. If you have deposited data etc in the Dryad repository, please include both the 'For publication' link and 'For review' link at this stage.
- If you are requesting an article processing charge waiver, you must select the relevant waiver option (if requesting a discretionary waiver, the form should have been uploaded at Step 3 'File upload' above).
- If you have uploaded ESM files, please ensure you follow the guidance at <https://royalsociety.org/journals/authors/author-guidelines/#supplementary-material> to include a suitable title and informative caption. An example of appropriate titling and captioning may be found at https://figshare.com/articles/Table_S2_from_Is_there_a_trade-off_between_peak_performance_and_performance_breadth_across_temperatures_for_aerobic_scooping_in_teleost_fishes_/3843624.

Author's Response to Decision Letter for (RSOS-201835.R0)

See Appendix B.

RSOS-201835.R1 (Revision)

Review form: Reviewer 2

Is the manuscript scientifically sound in its present form?

Yes

Are the interpretations and conclusions justified by the results?

Yes

Is the language acceptable?

Yes

Do you have any ethical concerns with this paper?

No

Have you any concerns about statistical analyses in this paper?

Yes

Recommendation?

Accept as is

Comments to the Author(s)

Thank you for your response. You have made good changes to your manuscript to address most of my concerns. While sequencing of qPCR amplicons was not pursued, I understand that it is often logistically difficult and/or impossible to do so for a project, especially if the experiments were completed multiple years ago. Even without any taxonomic data from the qPCR amplicons, the story presented in this manuscript is still clear and compelling. Nice work.

Decision letter (RSOS-201835.R1)

Dear Dr Tilstra,

It is a pleasure to accept your manuscript entitled "Relative abundance of nitrogen cycling microbes in coral holobionts reflects environmental nitrate availability" in its current form for publication in Royal Society Open Science. The comments of the reviewer(s) who reviewed your manuscript are included at the foot of this letter.

You can expect to receive a proof of your article in the near future. Please contact the editorial office (openscience@royalsociety.org) and the production office (openscience_proofs@royalsociety.org) to let us know if you are likely to be away from e-mail contact – if you are going to be away, please nominate a co-author (if available) to manage the proofing process, and ensure they are copied into your email to the journal.

on behalf of Pete Smith (Subject Editor)
openscience@royalsociety.org

Reviewer comments to Author:
Reviewer: 2

Comments to the Author(s)

Thank you for your response. You have made good changes to your manuscript to address most of my concerns. While sequencing of qPCR amplicons was not pursued, I understand that it is often logistically difficult and/or impossible to do so for a project, especially if the experiments were completed multiple years ago. Even without any taxonomic data from the qPCR amplicons, the story presented in this manuscript is still clear and compelling. Nice work.

Appendix A

Summary:

In this study, the authors seek to test the hypothesis that prokaryotic members of the coral holobiont regulate holobiont Nitrogen levels to maintain a stable symbiosis between the coral host and Symbiodinaceae and a healthy coral holobiont overall. Specifically, the authors sought to observe if the relative strength of denitrification vs. nitrogen fixation shifted in response to seasonal changes in Nitrogen availability (along with other parameters), and if this did or did not have an effect on Symbiodiniaceae cell counts in the coral holobiont. A coral reef in the Red Sea was surveyed seasonally for a variety of environmental parameters including DIN, DOC, DO, PAR, temperature, etc. Corals were also sampled at the same time scale, Symbiodiniaceae counts were taken, and qPCR was used to assess the relative ratio of nirS to nifH gene abundance as a proxy for the relative influence of denitrification and nitrogen-fixation. The effect of season on nirS:nifH ratios and Symbiodiniaceae cell densities as well as correlations between environmental parameters and these response variables were then tested.

Broadly, the authors found seasonal fluctuations in the relative nirS:nifH ratios and Symbiodinaceae cell counts. Correlations with environmental parameters demonstrated that nirS:nifH and Symbiodinaceae cell counts are positively correlated with environmental Nitrate availability. The authors conclude that a) seasonal fluctuations in nirS:nifH and Symbiodinaceae counts are driven primarily by Nitrate availability, b) increasing nirS:nifH in response to increasing Nitrate reflects beneficial behavior of prokaryotes in the coral holobiont, which could dampen fluctuations of Nitrogen in the coral relative to in the environment via changes in the relative strength of denitrification vs. nitrogen fixation, but c) this still was not enough to fully stabilize Symbiodinaceae populations.

Assessment:

This manuscript presents a fairly straightforward story on the role of nitrogen cycling prokaryotes in the coral holobiont. The order-of-magnitude changes in nirS:nifH fluctuations between seasons are compelling and the conclusion that this is driven by Nitrate availability makes sense. However, due to the nature of the data the paper can only make limited conclusions about what is happening within the coral holobiont. Specifically, a) qPCR only reflects gene copy number not necessarily rates of processes, b) no real conclusion can be drawn about how “beneficial” these changes may or may not be, and c) the actual availability of Nitrogen within the holobiont is never explicitly measured. Because of these gaps, I suggest the authors include additional data to strengthen their story.

Specifically, I think it would substantially bolster the paper to provide some information on which prokaryotes are responsible for the observed changes in nirS:nifH. The most straightforward way to do this would be to sequence the qPCR amplicons used in this paper. I recommend including this information in the next draft of the manuscript.

For the above reasons, I recommend major revisions to this paper prior to resubmission.

Line Comments:

52: Change from “including their” to “including by their”

57: Clarify to the reader that photosynthesis is done by Symbiodinaceae in the coral holobiont.

112: qPCR samples were run in technical replicates of 3. Were these averaged for each biological replicate prior to statistical analysis? Make this clear, because statistics should not be run on the technical replicates.

157-159: Where in the water column were the “sea water samples” taken?

172-173: Why were PERMANOVAs used? It is my understanding that two PERMANOVAs were run, one with nirS:nifH as the response variable and one with Symbiodiniaceae cell densities as the response variable (Table S1). In both cases, Season and Coral Species were the two predictor variables. If this is right then PERMANOVAs are being used on univariate data, which is an incorrect use of PERMANOVA since the response variable should be a distance matrix generated from multivariate data. Another type of model should be used (lm, glm, mixed mod, etc) that deals with univariate response variables (either normal or non-normal). The same holds true for all PERMANOVAs run on univariate environmental parameters. Additionally, individual identities of corals have been shown to influence microbiome community structure (along with other parameters), so any model that is run should use either a fixed or random effect to deal with the effect of “host ID”/“host genotype.”

268: If both nirS:nifH and Symbiodiniaceae cell densities are correlated with nitrate, why are they not be correlated with each other? Any ideas? There should be another panel in Figure 3 showing symb vs. nirS:nifH to visually show that they are not correlated.

289: Change “the here presented approach” to “this approach”

295-296: Some mention should be made of other processes in the N cycle that might be important in the coral holobiont.

323-325: The clause “this suggests that environmental N availability was closely linked with N availability within the coral holobiont in the present study as previously observed in ex situ studies” is crucial for your paper. N availability here is being measured in the water column, yet conclusions are being drawn about N availability in the coral holobiont. This implicitly assumes that N levels in water column reflect N levels in coral holobiont (at least before prokaryotic N cycling activities). You need to expand this clause to at least an additional sentence to really emphasize this to the reader.

338-340: “In this light, the observed increase in relative nirS to nifH gene abundance ratios with increasing N availability likely reflects a beneficial role of N cycling microbes in regulating N availability within the holobiont.” This does not “likely” reflect anything. It is one possibility, but a more straightforward explanation is simply that changes in nifH:nirS simply reflect the changing availability of the substrates acted on by these processes in the coral holobiont. Change the above sentence to be more measured in its conclusions and read something more like “relative nirS to nifH gene abundance ratios shift in response to the availability of environmental N, which could prove beneficial to the coral holobiont by partly stabilizing N levels in the host relative to environmental fluctuations.”

Appendix B

Reviewer: 1

Comments to the Author(s)

The authors have presented an insightful and well-written manuscript which is almost ready for publication. I only have a list of minor issues, which mostly concern the statistical part of the paper.

Our response: Thank you for the positive assessment of our manuscript.

Methods:

Line 152: what is this conversion factor based on? I do not see sources. Is this based on the spectral light quality in situ?

Our response: We have added the following information to the ms:

“The conversion factor was obtained by inter-calibrating the lux readings (i.e., from the Onset HOBO Pendant) with data obtained from a parallel deployment of a PAR sensor (LI-COR LI-1500 quantum sensor) during 4 h of daylight. Both readings correlated ($r^2 = 0.91$) and the obtained conversion factor was 51.8.” (see lines 153-156).

Results:

Supplementary Table S1 lists no interactive effect of species and season on gene ratios. Thus, the authors can only resort to testing differences between seasons for both species combined (i.e. averaging the data from both species for each season). However, as shown in figure 1B, the authors have tested the means of the different seasons against each other per species. Splitting the data into both factors and doing pairwise comparisons like this can only be justified when the PERMANOVA first yields a significant interaction. I recommend testing the various seasons against one another without splitting the data into both species (i.e. performing a post-hoc test to follow up the significant main effect of season). Indeed, this is what the authors have done and described in lines 205 to 207. I would therefore change the pairwise comparisons in figure 1 to reflect these lines.

Our response: Thank you. We have adjusted the figure to show differences only per season. Accordingly, we changed the Results section to reflect these changes.

Lines 194 to 198 should be removed as these suggest an interactive effect, which is not there. I would first report significant effects and then report the ratios which significantly differ, with species pooled.

Our response: We argue that a short descriptive statement about the similar pattern of both species between seasons adds to the understanding of this figure. However, as the updated Figure 1 doesn't include differences between seasons within species anymore and based on the reviewers' comments concerning Table S2 we chose to delete lines 198-200 (from the old manuscript).

Supplementary table S2 should not list parts A and B, but should include C, where the species have been correctly pooled (and the rest of the table, of course). Interestingly, the authors did pool species for their correlations as described in lines 259 to 261. I agree with this approach, by the way.

Our response: Parts A and B have been deleted from the Table. C has been updated according to reviewer 2's comments.

I hope the pairwise tests account for inflation of the familywise error (i.e. the p-values were corrected to compensate for multiple comparisons. Was this why you did parallel Monte Carlo tests)? I would make this issue clear in the M and M, because p values can be adjusted or a Bonferroni correction (with α / number of pairwise tests) can be done. In relation to this: Figure 1C is correctly done as there is a significant interactive effect for cell densities.

Our response: Monte Carlo tests were indeed done in parallel to account for multiple comparisons. We've added this information to the manuscript, see lines 182-185.

Table S2 details pairwise comparisons for environmental variables (f to o), but I do not see a main PERMANOVA table with season as factor and a corresponding pseudo-F, which should be presented before performing pairwise (post-hoc) comparisons, correct? If so, table S1 should be expanded accordingly. Indeed, line 174 does mention one-way PERMANOVA's but these are not listed in table S2.

Our response: Table S1 has been expanded as suggested.

The axes of figure 2 are hard to read; can the font size be increased?

Our response: Figure 2 has been changed accordingly based on new statistics and suggestions from the reviewer.

Table S3 has a colour legend, it seems easier to read if you just write down the actual r values with asterisks for significance. Perhaps the editor can decide on this?

Our response: We have updated the Table S3 as suggested.

The r values shown in figure 3 and table s3 are significant, but only weak to moderate. I would reflect this in the text. Now, lines 262 to 264 could suggest to the reader that strong correlations have been found, which is not the case.

Our response: Changed as suggested (lines 270 and 272).

Discussion:

Line 301:, regardless of species (there was no interactive effect).

Our response: Changed as suggested (line 312).

Line 311: ...showed a moderate correlation...

Our response: Changed as suggested (line 323).

I really like the discussion; perhaps the authors could also briefly touch upon the zoox densities found per se? The highest values, around 1 million zoox per cm², are what we commonly find in healthy corals growing in aquaria (where nutrient levels are quite higher than in the field). I am unsure about common densities found in the field, but perhaps it would be interesting to briefly mention something about other field data from the literature. Are these densities in a typical range for the field?

Our response: We have added an additional sentence to the Discussion where the densities were compared with densities of conspecifics originating from the Red Sea (see lines 333-334).

Reviewer #2

Summary: In this study, the authors seek to test the hypothesis that prokaryotic members of the coral holobiont regulate holobiont Nitrogen levels to maintain a stable symbiosis between the coral host and Symbiodinaceae and a healthy coral holobiont overall. Specifically, the authors sought to observe if the relative strength of denitrification vs. nitrogen fixation shifted in response to seasonal changes in Nitrogen availability (along with other parameters), and if this did or did not have an effect on Symbiodiniaceae cell counts in the coral holobiont. A coral reef in the Red Sea was surveyed seasonally for a variety of environmental parameters including DIN, DOC, DO, PAR, temperature, etc. Corals were also sampled at the same time scale, Symbiodiniaceae counts were taken, and qPCR was used to assess the relative ratio of *nirS* to *nifH* gene abundance as a proxy for the relative influence of denitrification and nitrogen-fixation. The effect of season on *nirS*:*nifH* ratios and Symbiodiniaceae cell densities as well as correlations between environmental parameters and these response variables were then tested. Broadly, the authors found seasonal fluctuations in the relative *nirS*:*nifH* ratios and Symbiodiniaceae cell counts. Correlations with environmental parameters demonstrated that *nirS*:*nifH* and Symbiodiniaceae cell counts are positively correlated with environmental Nitrate availability. The authors conclude that a) seasonal fluctuations in *nirS*:*nifH* and Symbiodiniaceae counts are driven primarily by Nitrate availability, b) increasing *nirS*:*nifH* in response to increasing Nitrate reflects beneficial behavior of prokaryotes in the coral holobiont, which could dampen fluctuations of Nitrogen in the coral relative to in the environment via changes in the relative strength of denitrification vs. nitrogen fixation, but c) this still was not enough to fully stabilize Symbiodiniaceae populations. Assessment: This manuscript presents a fairly straightforward story on the role of nitrogen cycling prokaryotes in the coral holobiont. The order-of-magnitude changes in *nirS*:*nifH* fluctuations between seasons are compelling and the conclusion that this is driven by Nitrate availability makes sense.

Our response: We thank the reviewer for his/her in depth review of our manuscript.

However, due to the nature of the data the paper can only make limited conclusions about what is happening within the coral holobiont. Specifically, a) qPCR only reflects gene copy number not necessarily rates of processes

Our response: We agree with the reviewer that qPCR measurements on the genomic DNA level do not allow for a direct qualitative or quantitative assessment of biological processes. However, in two previous studies using similar techniques, species and environmental settings, we showed that qPCR gene abundances as presented here aligned with rates of both N₂ fixation (Pogoreutz et al., 2017 Frontiers) and denitrification (Tilstra et al., 2019 SciRep) rates. As such, we are confident that our data is a good proxy for actual rates. We changed the Discussion accordingly (see lines 312-318):

“While relative gene abundances of both marker genes (i.e. *nirS* and *nifH*) do not allow any direct conclusion regarding activities of associated biological processes, previous studies using corals from the same location showed that relative abundances of marker genes correlated with denitrification and N₂ fixation rates under these environmental conditions (Pogoreutz et al. 2017; Tilstra et al. 2019). Consequently, the observed patterns of *nirS* to *nifH* gene abundance ratios may translate into similar seasonal patterns for associated denitrification to N₂ fixation activities.”

and the potential problem using it as a proxy in lines 360-362:

“If indeed translatable to corresponding prokaryotic activity, the observed dynamics in functional N cycling gene abundance ratios may, thus, directly support coral holobiont functioning (Rädecker et al. 2015).”

b) no real conclusion can be drawn about how “beneficial” these changes may or may not be

Our response: We agree with the reviewer. Thus, we changed the wording of one of our main conclusions as suggested by the reviewer (see lines 354-356).

, and c) the actual availability of Nitrogen within the holobiont is never explicitly measured.

Our response: The reviewer is correct in stating that N within the holobiont is not measured in our study. However, we have incorporated the need for such measurements when we detail recommendations for future research (including the quantification of N) in lines 375-379.

Because of these gaps, I suggest the authors include additional data to strengthen their story. Specifically, I think it would substantially bolster the paper to provide some information on which prokaryotes are responsible for the observed changes in nirS:nifH. The most straightforward way to do this would be to sequence the qPCR amplicons used in this paper. I recommend including this information in the next draft of the manuscript. For the above reasons, I recommend major revisions to this paper prior to resubmission.

Our response: We certainly agree with the reviewer that these data would strengthen our story. Unfortunately, this research was done more than 3 years ago and it is logistically not possible to perform the suggested additional analyses. Importantly, at present the taxonomic identity of N cycling microbes in the coral holobiont as well as their respective activity remains largely unexplored. Hence, any sequencing of amplicons may provide a taxonomic perspective. But this unfortunately does not allow any new or different conclusions regarding the ecological consequences of the observed shifts in qPCR gene abundances for the coral holobiont. Hence, we are confident that the conclusions drawn in the present manuscript are not confounded by the lack of sequencing data. Our results provide a meaningful insight into the putative equilibrium between coral-associated N cycling microbes and prevailing environmental conditions.

Based on the findings presented in this manuscript, new research questions and more functional experiments can be developed. Therefore, we thank the reviewer for his/her valuable recommendations to improve follow up research. We added the suggestion regarding the need to identify N cycling prokaryotes to the final paragraph of our manuscript, see line 377.

Line Comments:

52: Change from “including their” to “including by their”

Our response: Changed as suggested.

57: Clarify to the reader that photosynthesis is done by Symbiodinaceae in the coral holobiont.

Our response: Clarified as suggested.

112: qPCR samples were run in technical replicates of 3. Were these averaged for each biological replicate prior to statistical analysis? Make this clear, because statistics should not be run on the technical replicates.

Our response: Made clear as suggested.

157-159: Where in the water column were the “sea water samples” taken?

Our response: Samples were taken from directly above the reef, this information is added to line 162.

172-173: Why were PERMANOVAs used? It is my understanding that two PERMANOVAs were run, one with nirS:nifH as the response variable and one with Symbiodiniaceae cell densities as the response variable (Table S1). In both cases, Season and Coral Species were the two predictor variables. If this is right then PERMANOVAs are being used on univariate data, which is an incorrect use of PERMANOVA since the response variable should be a distance matrix generated from multivariate data. Another type of model should be used (lm, glm, mixed mod, etc) that deals with univariate response variables (either normal or non-normal). The same holds true for all PERMANOVAs run on univariate environmental parameters. Additionally, individual identities of corals have been shown to influence microbiome community structure (along with other parameters), so any model that is run should use either a fixed or random effect to deal with the effect of “host ID”/“host genotype.”

Our response: According to Anderson 2017 (Wiley StatsRef: Statistics Reference Online), PERMANOVAs can be used on a one response variable. However, where we generated a matrix based on Bray-Curtis similarities, we should have based it on Euclidian distances. Based on the remarks from both reviewer 1 and reviewer 2 we redid the statistical analyses of the qPCR and zoox densities. We changed the description of statistical analyses accordingly and added Anderson 2017 to the list of references, see lines 177-182.

268: If both nirS:nifH and Symbiodiniaceae cell densities are correlated with nitrate, why are they not be correlated with each other? Any ideas? There should be another panel in Figure 3 showing symb vs. nirS:nifH to visually show that they are not correlated.

Our response: It is likely that the high variability in the nirS:nifH data in combination with variability within the cell density data caused the lack of significant correlations. Extra panels with non-significant correlations have been added to Figure 3.

289: Change “the here presented approach” to “this approach”

Our response: Changed as suggested.

295-296: Some mention should be made of other processes in the N cycle that might be important in the coral holobiont.

Our response: We added a sentence about other potentially important N cyclers in the coral holobiont to the final paragraph of the Discussion, see lines 372-373.

323-325: The clause “this suggests that environmental N availability was closely linked with N availability within the coral holobiont in the present study as previously observed in ex situ studies” is crucial for your paper. N availability here is being measured in the water column, yet conclusions are being drawn about N availability in the coral holobiont. This implicitly assumes that N levels in water column reflect N levels in coral holobiont (at least before prokaryotic N cycling activities). You need to expand this clause to at least an additional sentence to really emphasize this to the reader.

Our response: We added a sentence highlighting the fact that internal N concentrations were not measured to the Discussion (lines 335-337).

338-340: “In this light, the observed increase in relative *nirS* to *nifH* gene abundance ratios with increasing N availability likely reflects a beneficial role of N cycling microbes in regulating N availability within the holobiont.” This does not “likely” reflect anything. It is one possibility, but a more straightforward explanation is simply that changes in *nifH*:*nirS* simply reflect the changing availability of the substrates acted on by these processes in the coral holobiont. Change the above sentence to be more measured in its conclusions and read something more like “relative *nirS* to *nifH* gene abundance ratios shift in response to the availability of environmental N, which could prove beneficial to the coral holobiont by partly stabilizing N levels in the host relative to environmental fluctuations.”

Our response: Changed as suggested (lines 354-356).

References

- Anderson MJ. 2017. Permutational Multivariate Analysis of Variance (PERMANOVA). Wiley StatsRef Stat Ref Online.:1–15. doi:10.1002/9781118445112.stat07841.
- Pogoreutz C, Rådecker N, Cárdenas A, Gärdes A, Wild C, Voolstra CR. 2017. Nitrogen fixation aligns with *nifH* abundance and expression in two coral trophic functional groups. *Front Microbiol.* 8:1187. doi:10.3389/fmicb.2017.01187.
- Tilstra A, El-Khaled YC, Roth F, Rådecker N, Pogoreutz C, Voolstra CR, Wild C. 2019. Denitrification aligns with N₂ fixation in Red Sea corals. *Sci Rep.* 9:19460. doi:10.1038/s41598-019-55408-z. www.nature.com/scientificreports.